# Reactive Flame-Retardant Cotton Fabric Coating: Combustion Behavior, Durability, and Enhanced Retardant Mechanism with Ion Transfer

**DOI:** 10.3390/nano12224048

**Published:** 2022-11-17

**Authors:** Wenju Zhu, Qing Wang, Mingyang Yang, Minjing Li, Chunming Zheng, Dongxiang Li, Xiaohan Zhang, Bowen Cheng, Zhao Dai

**Affiliations:** 1Tianjin Key Laboratory of Green Chemical Technology and Process Engineering, State Key Laboratory of Separation Membrane and Membrane Processes, School of Chemical Engineering, Tiangong University, Tianjin 300387, China; 2College of Chemistry Engineering & Materials Science, Tianjin University Science & Technology, Tianjin 300457, China

**Keywords:** cotton, flame retardant, thermal stability, DTPMPA, group synergistic

## Abstract

In recent years, we have witnessed numerous indoor fires caused by the flammable properties of cotton. Flame-retardant cotton deserves our attention. A novel boric acid and diethylenetriaminepenta (methylene-phosphonic acid) (DTPMPA) ammonium salt-based chelating coordination flame retardant (BDA) was successfully prepared for cotton fabrics, and a related retardant mechanism with ion transfer was investigated. BDA can form a stable chemical and coordination bond on the surface of cotton fibers by a simple three-curing finishing process. The limiting oxygen index (LOI) value of BDA-90 increased to 36.1%, and the LOI value of cotton fabric became 30.3% after 50 laundering cycles (LCs) and exhibited excellent durable flame retardancy. Fourier-transform infrared (FTIR) spectroscopy, X-ray photoelectron spectroscopy (XPS), and scanning electron microscopy (SEM) methods were used to observe the bonding mode and morphology of BDA on cotton fibers. A synergistic flame-retardant mechanism of condensed and gas phases was concluded from thermogravimetry (TG), cone calorimeter tests, and TG-FTIR. The test results of whiteness and tensile strength showed that the physical properties of BDA-treated cotton fabric were well maintained.

## 1. Introduction

With excellent hydrophilicity and biodegradability, cotton is widely used in the textile field and is recognized as a significant natural textile. While experiencing the convenience brought by cotton fabrics, people also suffer from the fire hazard caused by the flammability of cellulose fabrics. Improving the flame-retardant property of cotton fabrics has been for decades one of the problems to be solved. Halogen-free flame retardants have become a trend in recent years. Among halogen-free flame retardants, phosphorus-based flame retardants will become the leading research direction in the future [1,2]. Phosphorus-based flame retardants have good thermal stability and do not produce corrosive gases during pyrolysis, embodying good environmental friendliness. Numerous scientists with decades of research proved that these types of products do better with the addition of nitrogen. However, they still cannot compete with the traditional halogen-based flame retardant. Therefore, further improving the efficiency of phosphorus-based flame retardants is one of the hotspots in future research.

Boron is a member of the elemental semiconductor group with properties intermediate between metals and nonmetals [3]. Borate is one of the most widely used flame retardants in cellulose-based materials [4]. It can promote polymer dehydration and char formation by forming a glassy coating on the surface when exposed to fire [5]. The inhibition mechanism is similar to that of Al(OH)_3_. In addition, boron can be introduced into the polymer structure skeleton by designing an organic synthesis route. Boric acid applies a perfect combination of surface and organic chemistry to cotton fabrics [6,7]. Researchers have made many efforts to develop green, innovative, and promising high-efficiency, boron-based flame retardants. Through several studies and demonstrations, it was found that synergistic boron systems with other flame-retardant elements could further improve the efficiency of boron-based flame retardants. Systems such as phosphorus–boron [8], silicon–boron [9], phosphorus–silicon–boron [10] and phosphorus–nitrogen–boron [11,12] were applied to various flammable polymers.

However, the application of this boron-containing multielement synergistic system to cotton fabrics is rarely reported. Our group is committed to researching high-efficiency, halogen-free, and formaldehyde-free environment-friendly flame retardants to realize high-efficiency flame retardants in cotton fabrics. We put forward the idea of phosphorus–nitrogen–boron synergistic flame retardant. Through phytic acid coordination, the efficient and durable flame retardancy of phosphorus–nitrogen–boron synergistic flame retardant on cotton fabric was improved [13,14]. It is concluded that the treated samples maintain good flame retardancy and thermal stability after multiple washing cycles. FTIR and XPS analysis showed that boron existed on the surface of cotton fabrics in the form of B–O–C bonds, which indicated its excellent durability.

The feasibility of treating the cotton fabric with phosphorus–nitrogen–boron elements is reflected in the following aspects: (i) The boron atom with empty orbitals can coordinate with electron-rich nitrogen atoms to form stable covalent bonds [15]. (ii) The chemical combination of boron and phosphate can reduce the acidity of phosphoric acid in flame retardants and reduce the influence on the physical properties of cotton fabric during finishing. (iii) Boron-based flame retardant is easy to hydrolyze, and the chemical combination of boron and phosphate not only improves the stability of boron but also reduces the degradation rate of cotton fiber in the initial heating stage catalyzed by acid and improves the thermal stability of the material. (iv) The catalytic activity of boric acid further optimizes the connection between phases [16], and it is beneficial in reducing the influence on the physical properties of cotton fabric. It has become one of the key research directions for us further to verify the feasibility of phosphorus–nitrogen–boron synergistic flame retardant and to develop a low-cost, chelating coordinated phosphorus–nitrogen–boron synergistic flame retardant suitable for large-scale preparation.

Polyphosphonates have strong metal chelating ability to remigrate heavy metals from sediments and treated sludge, and they are widely used in water treatment, such as in oxide stabilizers, corrosion inhibitors, scale inhibitors, degreasers, and detergent additives [17,18,19,20,21]. As the most common amino polyphosphonate, DTPMPA has five phosphate groups in its structure, imparting a powerful chelating ability. It is usually used as the first choice of chelating agent since some of its complexes are precipitation-free in a wide pH range. DTPMPA has the characteristics of biodegradability, sustainability, low toxicity and relatively low cost [22]. The phosphonic acid group of DTPMPA is easy to connect with the organic part, which is conducive to synthesizing many different ligands [23]. DTPMPA has high phosphorus content (~22%) and a C-P bond, which makes it resistant to hydrolysis and thermal decomposition. These properties indicate that DTPMPA has the potential to be a “green” flame retardant finishing agent for fire-resistant and washing-resistant cellulose substrates, and it has been proven to have sound effects in many applications. For example, Jin et al. [24] prepared a new monomolecular intumescent flame retardant (DTM) by self-assembly DTPMPA and melamine. The UL-94 grade of DTM-modified polylactic acid PLA reaches the V-0 grade. Zheng et al. [25] synthesized a semi-durable flame retardant (ADTPMPA) for cotton fabric with DTTPMPA and urea. The 90 g/L ADTPMPA grafting treatment increased the LOI of cotton to 41.5%, and the cotton fabric’s tensile strength and flexibility were well retained.

A novel phosphorus–nitrogen–boron synergistic chelating flame retardant (BDA) is reported in this contribution. A new, friendly flame retardant with good heat resistance, low volatility, and long-lasting flame retardancy was developed by one-pot synthesis using the persistent ligand chelation of DTPMPA with boron. The BDA provided long-lasting functionalization of flame-retardant-modified cotton fabrics. We investigated the effect of BDA on cotton fabrics’ thermal stability and fire resistance and proposed a synergistic flame retardant and smoke suppression mechanism for cotton fabrics treated with BDA.

## 2. Materials and Methods

### 2.1. Materials 

Cotton fabrics (156.2 g/m^2^) originated from Dahutong Market in Tianjin, China. The surface was washed with 40 g/L NaOH solution at 70 °C for 0.5 h before application. After the above process, the surface was cleaned with deionized water. Diethylenetriaminepenta (methylene-phosphonic acid) (DTPMPA) (50 wt.% aqueous solutions) was supplied by Zaozhuang Kairui Chemical Co., Ltd. (Zaozhuang, China). Dicyandiamide, boric acid, urea, and ammonium phosphate monobasic were purchased from Aladdin Reagent Co., Ltd. (Shanghai, China). All chemicals were of reagent grade without further purification.

### 2.2. Synthesis of BDA 

Figure 1a shows the synthetic route of BDA. DTPMPA (16.98 mL, 0.02 mol, 50 wt.% aqueous solutions) and boric acid (2.55 g, 0.04 mol) were added into a 250 mL three-necked flask equipped with a mechanical stirrer. The mixture was stirred at 130 °C for 8 h, then cooled down to 30 °C. After the addition of urea (12.01 g, 0.20 mol) and deionized water (20 mL, 1.11 mol), it was reacted for another 4 h at 100 °C. The BDA was obtained as a maroon viscous liquid and was used without further purification. Using DTPMPA as a backbone, boric acid randomly combines with DTPMPA to form a new chemical bond, at which time a large amount of H^+^ is present in the solution. After adding urea and deionized water, an ion transfer phenomenon occurs. –NH_2_ replaces –OH on DTPMPA or gets H^+^ on the surface to generate NH_4_^+^.

### 2.3. Flame-Retardant Treatment of Cotton Fabrics

The flame retardant finishing solution was prepared by using BDA (30, 60, 90, and 120 g/L) as solute and deionized water as solvent. These solutions contained 40 g/L dicyandiamides, 40 g/L dicyandiamide phosphate, and 20 g/L urea as catalysts.

Impregnation of cotton fabrics with BDA finishing solution achieves approximately 100% moisture absorption at a fabric-to-solution ratio of 1:20 (*w*/*w*). The fabric was ultrasonicated in the finishing solution at 50 °C for 0.5 h. The wet textiles were dried at 100 °C for 0.5 h and further cured at 170–180 °C for 5–15 min. This procedure was repeated 1–5 times for an excellent curing effect and durability. After washing (distilled water) and drying, the flame-retardant cotton fabrics were prepared and named BDA-30, BDA-60, BDA-90, and BDA-120, respectively.

A proposed model of BDA grafted on cotton fiber is shown in Figure 1b [26]. All samples were weighed before and after treatment, and the values were fitted to Equation (1) to obtain weight gain:Weight gain (%) = [(W_1_ − W_0_)/W_0_] × 100%(1)

W_0_ and W_1_ represent the weight of the cotton fabrics before and after BDA finishing, respectively.

### 2.4. Characterization 

All cotton samples were washed in accordance with the AATCC Monograph 6—2016 test method. The whole laundering process was carried out in a conventional washing station (Haier, EB65M019) of a 6.5 kg automatic washing machine. The washing environment was at 25 °C without any detergent addition. The duration of each laundering cycle was 15 ± 2 min.

FTIR spectra were acquired based on a NICOLET IS10 spectrometer (Thermo Fisher Scientific Co., Waltham, MA, USA) with technical specifications requiring a resolution of 0.5 cm^−1^ and a wave number range from 4000 to 400 cm^−1^.

XPS analysis was carried out on a K–αXPS spectrometer (Thermo Fisher Co., USA). Measurements conducted utilizing the standard test method ASTM D6413-08: setting a tube voltage at 36 kV, a tube current at 20 mA, and a scan step at 0.01. Before testing, the fiber samples must be dried. The samples were exposed to Cu Kα radiation at 40 °C for 2 h in a vacuum environment.

A scanning electron microscope (SEM, Phenom XL, Phenom China Co., Shanghai, China) was selected for the instrumentation. The operating environment is 15 kV beam voltage, 6~8 mm working distance. Scanning electron microscope (SEM) images of fabric samples were collected in this working environment.

Based on the ASTM D2863-2000 standard as the implementation guide, the LOI of the samples was determined by controlling the experimental conditions with a ZKHW-205 digital oxygen index meter (Qingdao Zhongke Hengwei Intelligent Technology Co., Ltd., Shandong, China). The sample needs to be dried in an oven at 80 °C for 4 h before the test. The gas pressure in the oxygen cylinder and nitrogen cylinder was not less than 5 MPa.

A vertical combustion flammability test was carried out following the standard GB/T 5455—2014. A YG815B vertical combustion tester (Nantong Sansi Electromechanical Technology Co., Ltd., Jiangsu, China) was selected for the study.

TG was determined with a DTG-60 thermogravimetric analyzer (Shimadzu Enterprise Management (China) Co., Ltd., Shanghai, China). The samples were heated from 30 to 900 °C using a heating rate of 10 °C/min, assisted by a 30 mL/min nitrogen flow rate. 

TG-FTIR analysis was performed with a STA449F3 + QMS403C + VERTEX80 (TG-MS-IR) spectrometer (Germany Netstal, Germany Bruker). Each sample (10 ± 0.2 mg) was tested at dynamic N_2_ (flow rate: 30 mL/min) at a temperature of 30–900 °C (heating rate: 10 °C/min).

The cone calorimeters test is based on a standard of ISO 5660-1 2015 under 50 kW/m^2^ of irradiative heat flux, using a model of TESTech-GBT 16172 conical calorimeter (Testech (Suzhou) Testing Instrument Technology Co., Ltd., Zhejiang, China). Square cotton samples (100 mm × 100 mm) were used. The baseline oxygen concentration was 20.81%, and carbon dioxide was 0.06%.

Using GB/T 3923.1-2013 as a guide, the tensile strength database was collected using an electronic universal tensile tester CMT4503 (MTS SYSTEMS (China) CO., Ltd. Shanghai, China).

The whiteness value results were obtained from a WSB-1 Portable Whiteness Meter (Shanghai Xuancheng Instruments Co., Ltd., Shanghai, China) The cotton fabric was folded into eight layers and the whiteness value was tested on a test bench. Each sample was conducted five times, taking the average value.

## 3. Results and Discussion

### 3.1. Finishing Conditions 

The cotton fabrics were treated following the experimental procedure and were repeatedly cured at 170–180 °C for 5–15 min. We integrated the weight gain data for fabrics treated at various curing times and laundering cycles. The results are shown in Figure 2. Obviously, the weight gain rate increased with curing times, and the fabric treated with BDA-90 had the highest weight gain, even more than that of the one treated with BDA-120, regardless of curing times (Figure 2a). This is mainly because the BDA can be uniformly grafted on the surface of the cotton fabric at a concentration of 90 g/L. When its concentration was high, the binding fastness to the cotton fabric would decrease due to the formation of chelation coordination between the BDA flame retardants, and these self-coordination FRs cannot sustain violent washing. Because the weight gain rate has a positive correlation with flame retardancy, to minimize the strength loss and maximize the weight gain rate, three curing times were the best choice [14]. The weight gain (three curing times) and laundry cycle results (Figure 2b) reveal that the weight gain rate gradually decreased with the increase of laundering cycles, and the weight losses were reduced after 30 times. The total weight loss of the samples after 50 laundry cycles was calculated at about 35–45%. Therefore, 90 g/L BDA concentration and three curing times were selected as the best procedure for the following experiments.

### 3.2. FTIR Spectra Analysis 

The BDA, control sample, and BDA-treated samples (3 curing times, 50 LCs) were analyzed via FTIR. The results are illustrated in Figure 3. The absorption peaks of BDA are clear and easily recognized. The spectral bands from 3200–3700 cm^−1^ and the peak at 1600 cm^−1^ correspond to the tensile and bending vibrations of H-O-H, respectively [27,28]. It indicated the presence of slight coordination water in BDA, which is consistent with the results of BDA decomposition from 103.7 °C in TG data. The peak at 2850 cm^−1^ is associated with the C–H vibration absorption; the peaks at 1425 cm^−1^ and 1040 cm^−1^ are ascribed to the B–N [29] and B–O–C groups [30], respectively; the absorption peak at 1160 cm^−1^ is associated with the P=O vibration, and the weak peaks at 910 cm^−1^ are assigned to the P–O–H vibration absorptions. As regards the control cotton, the peaks at 3300 cm^−1^ and 2890 cm^−1^ are attributed to the O–H and C–H vibration absorption, respectively [31]. The peaks at 1054 and 1030 cm^−1^ correspond to the C–O vibration absorptions [32]. Compared with the control samples, new absorption peaks of BDA-treated samples are weak and covered by the original signal peaks of control cotton fabric. The absorption peaks that appeared at 1440 cm^−1^ and 1315 cm^−1^ are ascribed to the B–N and B–O vibration absorption [33]; the peaks at 1030 cm^−1^ and 906 cm^−1^ are assigned to the P–O–C vibration absorptions, respectively [34]. The new peaks reveal the esterification reaction between C–OH and P–OH groups. Besides, the peak of 1710 cm^−1^ is attributed to the C=O stretching vibration [35], which may indicate the oxidation of C–O–C bonds of cellulose in the process of cure finishing [36].

### 3.3. XPS Analysis

To further explore the chemical composition, chemical state, and working mechanism of BDA on the cotton fiber surfaces, the XPS method was applied to detect the element contents in the control cotton, BDA-90-treated samples (3 curing times, 50 LCs), and the char residue. Figure 4a demonstrates to us survey spectra. Table 1 contains data on the chemical composition (at.%) of the XPS surface. Two firm peaks at 285.25 eV and 532.21 eV correspond to C1s and O1s of the control and the BDA-90 treated samples. However, for the BDA-90-treated sample, three novel peaks were detected in the XPS spectrum—400.09, 190.81, and 133.37 eV—coinciding with the characterized signals of N1s, B1s [29], and P2p [37,38,39] levels, respectively. Core levels of B1s in B(OH)_3_ were reported to range from 193.0–193.6 eV [40,41]. V.V. Atuchin et al. showed that the B1s’ binding energies of B in β-BaB_2_O_4_ and KRbAl_2_B_2_O_7_ borate crystals measured 191.6 and 191.8, correspondingly [42,43]. Owing to the presence of the sp^2^ and sp^3^ bonds of B–N, the B1s’ peak in Figure 4a appears at 190–191 eV [44]. Therefore, it indicated that the boron element in BDA chelates with nitrogen to form a durable flame retardant rather than a single borate ester on the surface of the cotton. After 50 LCs, the phosphorus, boron, and nitrogen element content on the cotton fiber surface were 1.75%, 2.67%, and 4.72%, respectively, as listed in Table 1. Noticeably, the elemental boron content on the surface of the treated samples exceeded the elemental phosphorus content by about 50%. The possible reason is that the phosphorus in the treated sample can better seep into the amorphous area of the cotton fabrics and accumulate on the fiber surface during combustion to play a dehydrated and carbonated role. In contrast, the boron element was mainly on the fiber surface due to chelating coordination and formed a dense barrier layer upon the fire to play a flame-retardant function. The XPS surface analysis technique concluded that the BDA promoted the flame-retardant process of the condensed phase, which was beneficial to the carbon formation of cellulose.

### 3.4. Flame Retardant Properties and Durability

The vertical flammability test was used to visualize the ignition resistance of control samples and samples containing BDA. The results are shown in Figure 5. After 12 s of ignition, the control samples burned out without leaving any residue. All BDA-treated samples were charred only in the area in direct contact with the flame, whereas the remaining samples were unaffected. The char length of BDA-30, BDA-60, BDA-90, and BDA-120 treated samples (3 curing times and 50 LCs) were 73 mm, 70 mm, 55 mm, and 63 mm, respectively. The results showed that the BDA-90-treated sample had the best flame retardancy, consistent with the oxygen index data and SEM image results.

The LOI values of fabric samples treated with different concentrations of BDA (3 curing times) after 1–50 washing cycles, respectively, are shown in Figure 4b and listed in Table 2. All LOI results were over 26.0% after 50 launderings, whereas the untreated cotton fabric had an LOI of ca. 18.4 %. As prepared, BDA-treated cotton fabric samples were initially processed for one washing cycle. The BDA-90 treated sample had the most extensive LOI data, 36.1%, and the BDA-60 and BDA-120 treated samples were similar at 33.6% and 33.7%, respectively. The decreased flame-retardant property with the increased BDA concentration may be caused by reduced interaction and durability of concentrated BDA on cotton fabrics, causing a higher weight loss rate of BDA during laundering. After ten laundering cycles, the LOI data of all samples were between 31.8—32.9%, which showed that their flame retardancy was very similar, indicating powerful interactions of the flame-retardant system with cotton fabrics at the concentrations of BDA in the range of 30–90 g/L. Although the concentration was increased by four times, the LOI data (after 50 laundering cycles) of BDA-90- and BDA-120-treated samples were 30.3% and 30.1%, respectively. And the BDA-60-treated sample shows 28.9%. The BDA-30-treated sample has the smallest at 27.8%. The results show that the current finishing procedure could produce the cotton fabric with a high grafting rate at a lower concentration of BDA.

A conical calorimeter was used to evaluate the combustible behavior of the control sample and the BDA-90-treated sample. The results are presented in Figure 6 and Table 3. The ignition time of the control cotton was the same as that of the BDA-90 finished sample (TTI = 9 s), which indicated that the BDA has a weak catalytic effect on shortening the ignition time [33]. The BDA-90 finished sample left 20.1% residue at the end of the test, whereas no residue was found in the control cotton group. The effective heat combustion (EHC), and total smoke rate (TSR) values of the BDA-90-treated sample were approximately 50% less than that of control cotton, whereas the heat release rate (HRR) of BDA-90-treated samples was 15.84 MJ/m^2^, significantly lower than that of the control sample (46.44 MJ/m^2^). Notably, the peak heat release rate (pkHRR) and the total heat release (THR) of BDA-treated cotton fabrics were 78.91 % and 50.52%, respectively, lower than the control cotton fabrics. Combustion efficiency can be evaluated by the ratio of CO_2_ to CO. The higher the [CO_2_]/[CO] ratio, the more adequate the combustion [45]. The [CO_2_]/[CO] ratio of the BDA-90 was 3.41, significantly lower than the primary fabric. It implied that BDA flame retardant significantly reduced combustion efficiency and effectively hindered the combustion of cotton fabrics. Cone calorimetry results show that the BDA-treated sample significantly inhibited the combustion process and reduced the release of heat and smoke, which implied a flame-retardant mechanism of condensed matter expansion.

The combustion of the BDA-functionalized cotton fabric is effectively controlled and provides continuous durability by LOI, vertical combustion, and cone calorimetry tests.

### 3.5. Thermal Stability

To analyze the effect of BDA on the stability of cotton fabrics, a pyrolysis survey was conducted. Figure 7 presented us the TG and DTG diagrams of cotton fabrics. Corresponding data can be found in Table 4. The TG and DTG curves of cotton fabrics in nitrogen are shown in Figure 7. The corresponding results are shown in Table 4. The mass loss process was divided into two stages: a: moisture and solvent removal step at T < 200 °C; b: thermal decomposition stage of the samples [46,47].

The thermal degradation of BDA was manifested in a three-stage weight loss process, as presented in Figure 7 and Table 4. Because the BDA was not purified before use, the sample began to lose weight in the range of 103.7 °C–147.6 °C, with a weight loss of 36.59% due to moisture and residual solvent evaporation. During the second stage of heating over 252.6–453.9 °C, decomposition of organic ligands occurred, with a mass loss of 21.14%, accompanied by the release of nonflammable gases [48]. Until heated to 900 °C, the residual char of the sample was 42.26%, presumably related to the char formation in the second stage.

From Figure 7a, a substantial reduction in mass was observed in the control cotton fabric after 200 °C, mainly occurring over 307.7 °C–409.1 °C. The maximum loss rate peak (T^b^ max) appeared at 343.2 °C, which corresponded to a maximum mass loss rate (R^b^ max) of 2.0414%/°C. This behavior represented the main pyrolysis phase characteristic of the control cotton fabrics. Small flammable molecules of gas were generated during this period. Eventually, the solid residue was 19.57% at 900 °C.

BDA modified the thermal decomposition of the treated cotton. As shown in Table 4, the initial pyrolysis temperature (T^a^ onset, T^b^ onset) and the peak maximum loss rate (T^a^ max, T^b^ max) were significantly lower in the treated cotton fabrics at stages a and b. Eventually, the carbon residue of BDA-90 in the 900 °C treated sample was 36.33%, which showed a considerable improvement over the control cotton. These effects could be explained by the fact that BDA degraded and released acids that promote dehydration and carbonization of cellulose [49]. In addition, there was disruption of BDA organic coordination bonds, accompanied by phosphonic acid generation. Phosphonic acid had a positive effect on the dehydration of cellulose into carbon. Cotton fabrics containing BDA had higher residual charring rates consistent with char formation theory.

### 3.6. TG-IR Analysis

To pursue a deeper exploration of the thermal degradation process of BDA-containing samples, TG-FTIR was performed to detect the real-time gaseous products released in the pyrolysis process and measure the composition changes at different temperatures (Figure 8). The FTIR data were plotted line by line to form a three-dimensional spectrum to reveal the differences in composition.

The substances vaporized during the pyrolysis of control cotton are visible, and the types of bonds corresponding to extraordinary absorption peaks are also listed in Figure 8a. Figure 8a showed the typical products released by thermal cracking from control cotton fabric were water vapor (~3400–3300 cm^−1^), hydrocarbons (~2990–2600 cm^−1^), CO_2_ (~2340–2320 cm^−1^ and 710 cm^−1^), CO (~2260–2120 cm^−1^), carbonyl compounds (~1850–1620 cm^−1^), and ethers (~1500–900 cm^−1^) [50]. The gases that decomposed and vaporized from the control cotton fabric (Figure 8a) were mainly water, CO_2_, and related carbon radicals, which come from oxidation, dehydration, decarbonylation, decarboxylation, and decomposition of cellulose under high temperature [51].

Nevertheless, the BDA-90-treated sample was different from this. Moreover, some new characteristic absorption peaks appeared. As can be seen by comparing the evolution profiles (Figure 8a,b), the peak ratio of CO_2_ (~2340–2320 cm^−1^) to CO (~2200–2120 cm^−1^) increased significantly, indicating that the carbon monoxide emissions were greatly reduced. The peak area at ~2990–2600 cm^−1^ was lower than that of control cotton, mainly due to the small proportion of released saturated hydrocarbons. In addition to water vapour, CO_2_, CO, hydrocarbon C–H, and C–C gases, some new characteristic peaks were found, mainly the signal peaks of phosphorus and nitrogen elements. In general, organophosphorus compounds may exhibit the absorption peak region of P(=O) OH and P–NH_2_ from 1750 to 1600 cm^−1^, absorption peaks of P–O–P asymmetric vibration and P-N vibration peaks at around 1040–1000 cm^−1^ and 1000–940 cm^−1^ [51]. Similarly, the absorption peaks of nitrogen -NH and NH_3_ were estimated to be at 3720–3450 cm^−1^ and 900–860 cm^−1^, respectively [52]. By comparing the TG-FTIR of the cotton fabric and the BDA treatment sample, the combustible carbon component in the gas phase of the BDA treatment sample was significantly reduced. Meanwhile, a series of nitrogen and phosphorus components were detected. In brief, it proved the validity and feasibility of gas-phase flame retardant.

### 3.7. Surface Morphology

Surface morphological modifications of the samples were visualized by SEM. The amplification and the consequences are shown in Figure 9. Based on Figure 9a–e, it can visually illustrated that the surface morphology of the BDA-treated cotton fabrics was without significant modifications. The effect of BDA concentration on morphology was negligible by comparing Figure 9b–e. The BDA-treated fibers were shrunken after combustion, and numerous particles appeared on the surface. The result suggested that the surface topography and structure of BDA-treated cotton fabrics were well maintained. After 50 washing cycles, BDA-treated cotton fabrics still showed good flame retardancy against combustion. Compared with the micrographs of the samples after combustion in Figure 9b’–e’, BDA-90 (d’) had the most particulates on the surface after combustion. The second were BDA-60 (c’) and BDA-120 (e’), while BDA-30 (b’) had the least particulates on the surface. The presence of particles proved the enrichment of BDA on the surface of cotton fabric when exposed to heat. During combustion, BDA pyrolysis occurred on the fiber surface, which promoted the charring of cellulose. The dense char layer hindered the mass and heat transfer among phases and the cellulose matrix was well protected [53]. The SEM images of the BDA-treated sample after the combustion are significantly different from those of the related literature reports without using boric acid [25].

### 3.8. Flame Retardant Mechanism

Combustion is a complex mechanical process, and the flame mechanism of cotton fabrics was widely investigated [54]. The consideration of the problem of flame-retardant cotton fabric was incomplete until now. The thermal degradation mechanism of polymers is directly correlated with their flame retardancy, and various types of flame retardants affect the thermal degradation process of cotton fabrics from several aspects, thus presenting a variety of flame-retardant mechanisms.

As polysaccharide macromolecular compounds, cotton fabrics contain abundant ether bonds (C–O–C) and methylene bonds (–CH_2_–) in the molecular chain. BDA contains three flame retardant elements: phosphorus, nitrogen and boron. The surface of BDA has a large number of N-containing groups, which is caused by enhanced ion transfer. NH_4_^+^ transfer to the surface of the flame retardant, the phosphorus–nitrogen synergistic flame retardant system is conducive to capturing oxygen free radicals(O•) and hydroxyl radicals (OH•) generated by the end-of-chain-reaction cracking in the combustion process [52]. TG-FTIR demonstrated the inference that the BDA flame retardant system involved the participation of the gas phase in flame retardancy (Figure 8).

According to the mentioned results, the possible thermal decomposition paths of the control cotton fabrics and treated cotton fabrics are shown in Figure 10 [55]. The control cotton fabric decomposed when heated and released a wide range of flammable volatiles such as hydrocarbons without melted drops and residual carbon during combustion. With the initial pyrolysis temperature, the Tonset and Tmax of the treated cotton fabric were slightly lower than the ordinary cotton fabric. The residue of the flame-retardant cotton fabric increased to 42.26% at 900 °C. The high carbon formation is attributed to the reaction of BDA, which influences the result of breaking of chemical bonds during the pyrolysis of cellulose. From the perspective of bonding energy, we could find that the chemical bonds present in the flame retardant system are listed from minor to major, as given in the Table 5 [56]. Before the pyrolysis of cellulose (103.7 °C–147.6 °C), the P-N, C-N and P-C bonds in BDA are broken and decomposed into intermediate polyphosphate chains (containing boronic acid groups) and phosphoric acid monomers, accompanied by the production of water, ammonia and other nonflammable gases. During the thermal degradation over 252.6 °C–453.9 °C, cellulose chains broke into hydroxyl-terminated fragments. The P-O bond is broken, and the polyphosphate group formed a preliminary char layer by esterification reaction with the terminal hydroxyl segment.

The value electron structure of boron is 2s^2^ 2p^1^. Boron atoms with empty orbitals tend to form coordination bonds with nitrogen atoms containing lone electron pairs [15]. B-N bond energy is as high as 389 kJ/mol, enhancing flame retardants’ thermal stability. The B-N coordination bond broke under heat to form a boronic acid derivative, catalyzing the cellulose decomposition rearrangement reaction, and the secondary amine derivative molecules were also involved in establishing the char layer. The molten state boric acid derivatives tended to migrate toward the char layer surface, forming a nonpermeable glazed layer on the surface of the phosphorus-based char. On the one hand, the transport of oxygen from the environment was prevented; on the other hand, the further volatilization of the pyrolysis products was prevented. The inference becomes convincing by the increase in surface element B’s relative content in the XPS analysis results.

Due to the B-O bonding energy being significantly higher than that of the C–O bond, the BDA-treated group of cotton fabrics exhibits a superior thermal stability under high temperatures in comparison with ordinary cotton fabrics [57]. The boric acid and phosphoric acid derivatives produced by the decomposition of BDA at 500–900 °C further catalyzed the cross-linking of alkene fragments generated by the thermal decomposition of cellulose to form a more extensive barrier char layer. This process decreased the release of combustible gases, allowed the pyrolysis products to remain in the condensed phase as much as possible, and formed a residual bulk carbon containing phosphorus and boron flame retardant elements with better expansion. The dense char layer contributed to the maintenance of the overall shape of the cohesive potential barrier, as shown in the SEM image (Figure 9).

Through deep discussion of the results, the flame retardant mechanism summarized is that gas phase and condensed phase cooperate to protect cotton fabrics.

### 3.9. Tensile Strength and Whiteness of Cotton Fabrics

As shown in Table 6, an electronic universal tensile testing machine measured the tensile strength of the control and the BDA-treated samples (3 curing times, 50 LCs). It is easy to see that the tensile strength’s loss rate increased with the increase of the BDA concentrations. The tensile strength of control cotton was 530 N, and that of the BDA-120- treated sample decreased to 393 N. However, even with the maximum concentration of BDA (120 g/L), the strength loss of treated samples was relatively moderate and acceptable.

The whiteness of the BDA-treated cotton fabrics and the relationships with different LCs are shown in Table 7. On the surface, there is no distinction between the BDA-containing samples and the routine samples. The data of whiteness and whiteness ratio (whiteness ratio, the ratio of whiteness of the BDA treated sample to the whiteness of the control sample multiplied by 100%, unit: %) showed that the whiteness of the sample gradually returned to the standard level (75.4) after washing. The data in Table 7 indicated that the concentration of BDA had a negative effect on the whiteness of the samples. Even so, the whiteness of BDA-120 was still widely acceptable. The boron element involved in organic coordination in BDA had a positive effect on the antiyellowing process of the fabric [58].

## 4. Conclusions

A novel ecofriendly flame retardant for cotton fabrics was successfully prepared. The experimental results show that DTPMPA ammonium salt could form a stable structure with boric acid, which enhanced the durability, while the ion transfer made the flame-retardant ability significantly improved. The significantly enhanced flame-retardant mechanism was explained by elaborating the combustion behavior, durability, and ion transfer. It could form a stable chemical bond on the surface of cotton fibers after repeating the curing processes three times. The formed BDA and treated fabrics were analyzed using FTIR spectra, XPS elemental analyses, and SEM micrographs. TG, LOI, and vertical burning tests analyzed the durability and flame retardancy. The cone calorimeter tests and TG-FTIR analysis concluded a synergistic, intumescent, flame-retardant mechanism. At the same time, the whiteness and tensile strength of the BDA-treated sample showed an appropriate effect. The novel BDA flame-retardant system provides an effective synergistic strategy for developing an efficient, durable, and ecofriendly flame retardant for cotton fibers. Our lab is exploring applying this synergistic flame-retardant system to polymeric materials such as polyesters, polyurethanes, and plastics.

## Figures and Tables

**Figure 1 nanomaterials-12-04048-f001:**
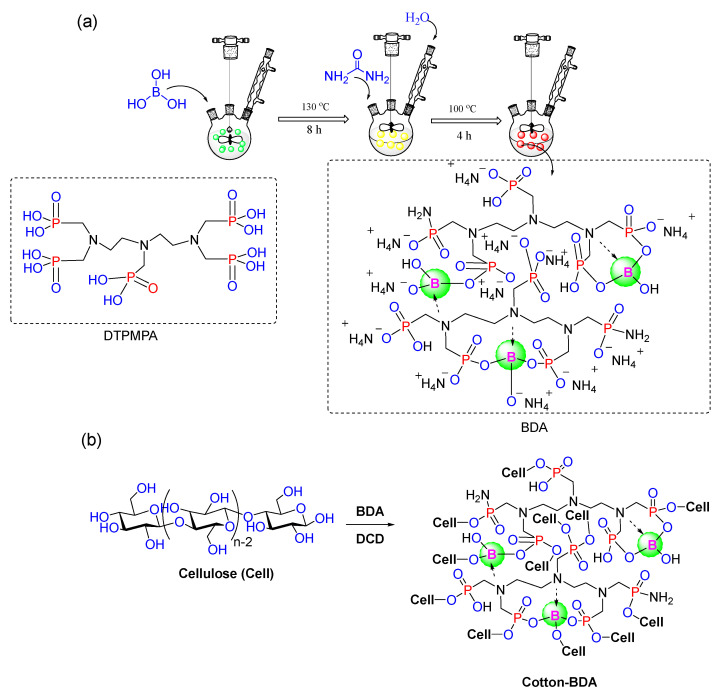
(**a**) Synthesis of BDA; (**b**) Proposed model of BDA interaction with cotton fiber.

**Figure 2 nanomaterials-12-04048-f002:**
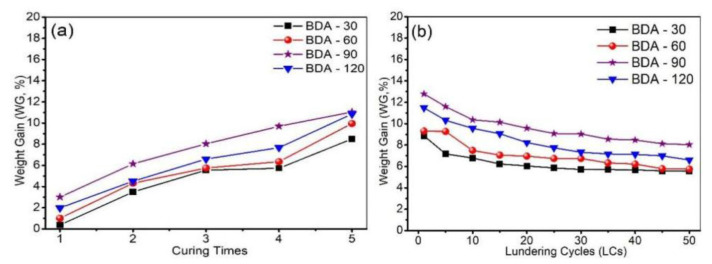
(**a**)The relationship between weight gain (50 LCs) and curing times; (**b**) the relationship between weight gain (curing times) and laundering cycles.

**Figure 3 nanomaterials-12-04048-f003:**
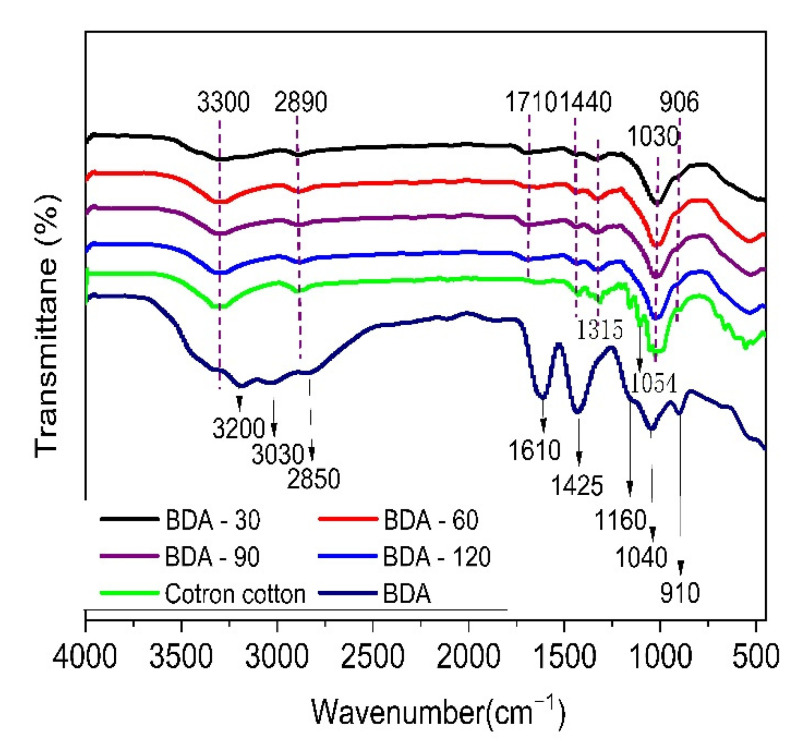
FTIR spectra of BDA, control sample, and BDA-treated samples (3 curing times, 50 LCs).

**Figure 4 nanomaterials-12-04048-f004:**
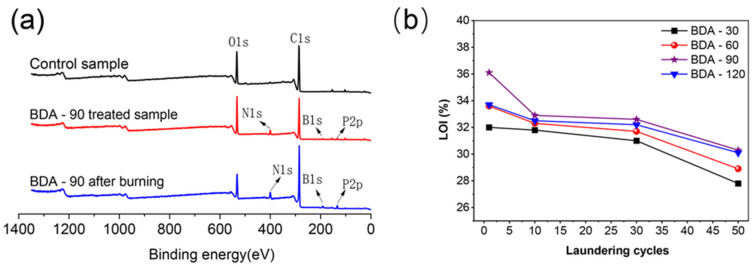
(**a**) XPS spectra of the control sample, BDA-90-treated sample (3 curing times, 50 LCs) before and after combustion; (**b**) LOI values of BDA treated samples (3 curing times) after 1–50 laundering cycles.

**Figure 5 nanomaterials-12-04048-f005:**
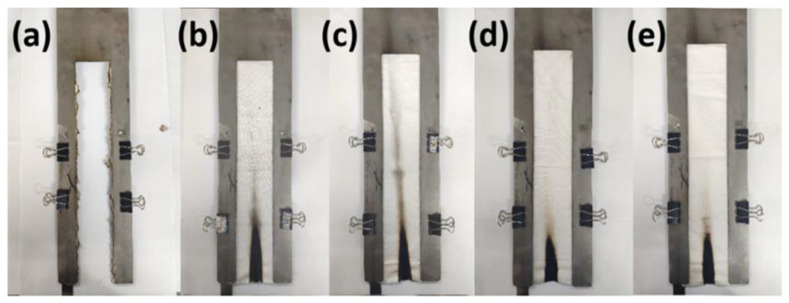
Photographs of vertical burning test of the control sample: (**a**) BDA-30; (**b**) BDA-60; (**c**) BDA-90; (**d**) BDA-120; (**e**) treated samples (3 curing time, 50 LCs).

**Figure 6 nanomaterials-12-04048-f006:**
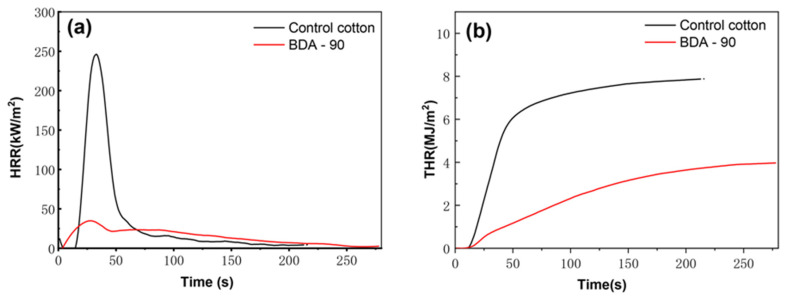
Heat release rate (HRR): (**a**) total heat release (THR); (**b**) curves of control sample and BDA treated samples.

**Figure 7 nanomaterials-12-04048-f007:**
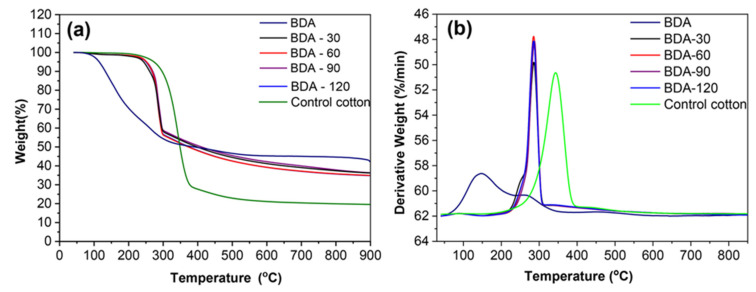
(**a**) TG and (**b**) DTG curves of BDA, control sample, and BDA-treated samples (3 curing times, 50 LCs).

**Figure 8 nanomaterials-12-04048-f008:**
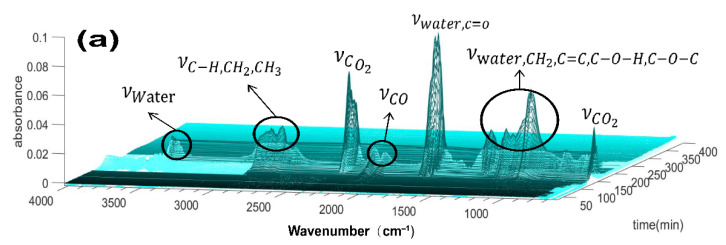
Three-dimensional TG-FTIR spectra of the control sample (**a**) and BDA-90-treated samples (**b**) (3 curing times, 50 LCs).

**Figure 9 nanomaterials-12-04048-f009:**
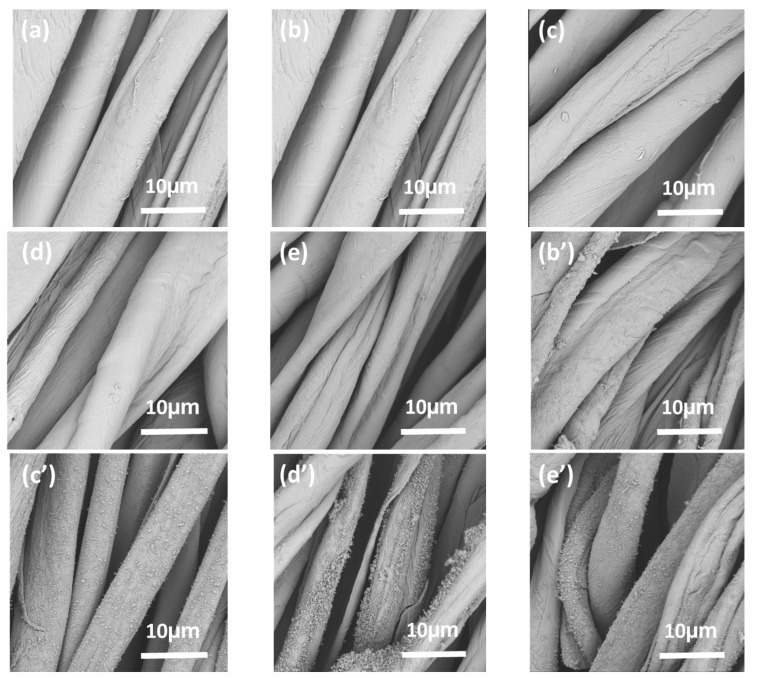
SEM micrographs of the control sample (**a**), BDA-30 (**b**), BDA-60 (**c**), BDA-90 (**d**), BDA-120 (**e**) (3 curing times, 50 LCs), and after burning (**b’**–**e’**), respectively. Scale bars are 10.0 µm (×5000).

**Figure 10 nanomaterials-12-04048-f010:**
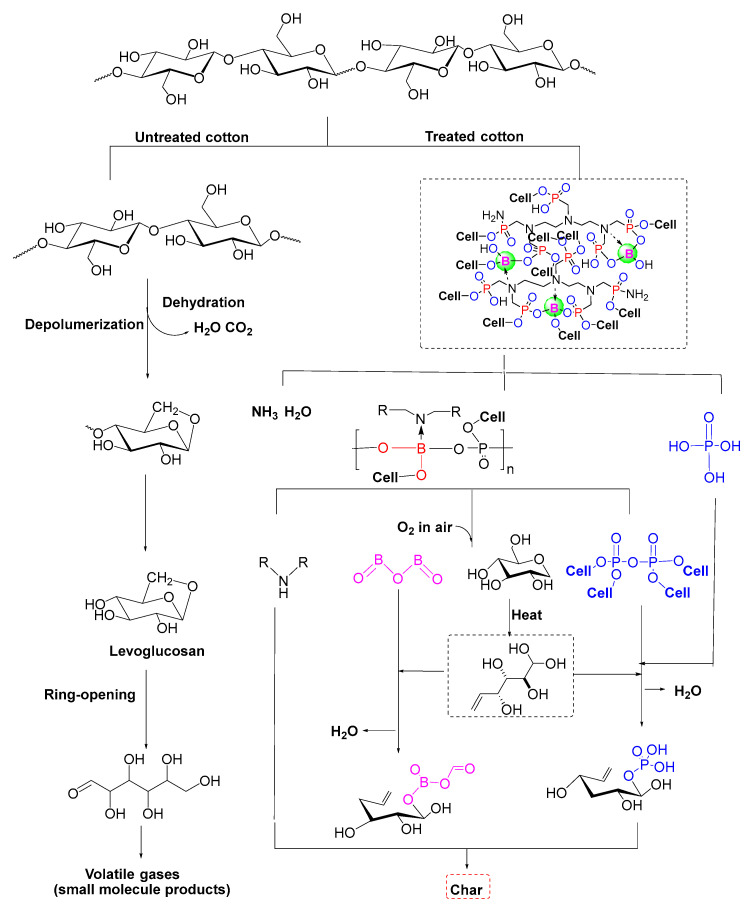
Possible pyrolysis routes of BDA-treated cotton fabrics.

**Table 1 nanomaterials-12-04048-t001:** XPS elemental analyses of the control sample, BDA-90 (3 curing times, 50 LCs) before and after combustion.

Sample	XPS Atomic Percent (at.%)				
	C	O	N	B	P
Control sample	73.45	21.11	-	-	-
BDA-90 treated sample	66.15	22.01	4.72	2.67	1.75
BDA-90 after burning	70.88	14.28	6.61	3.93	3.23

**Table 2 nanomaterials-12-04048-t002:** LOI data of BDA treated samples (3 curing times) after 1–50 laundering cycles.

Sample	TTI (s)	HRR (MJ/m^2^)	pkHRR (W/m^2^)	THR (MJ/m^2^)	EHC (MJ/kg)	TSR (m^2^/m^2^)	CO_2_/CO (kg/kg)	Residue (%)
Control sample (CV %)	9	46.44 (10.30)	206.2 (9.48)	7.76 (3.97)	19.3 (28.66)	113.0 (14.51)	41.41 (12.31)	-
BDA-90 (CV %)	9	15.84 (8.57)	43.50 (17.16)	3.84 (6.67)	10.2 (23.36)	67.3 (23.47)	3.41 (6.01)	20.1 (6.26)

**Table 3 nanomaterials-12-04048-t003:** The cone calorimeter test data of control sample and BDA-90-treated samples (3 curing times, 50 LCs).

Laundering Cycles	LOI (%)			
	BDA-30	BDA-60	BDA-90	BDA-120
1	32.0	33.6	36.1	33.7
10	31.8	32.3	32.9	32.5
30	31.0	31.7	32.6	32.2
50	27.8	28.9	30.3	30.1

**Table 4 nanomaterials-12-04048-t004:** TG analysis data of control sample and BDA treated samples (3 curing times, 50 LCs) under N_2._

Sample	T^a^ Onset (°C)	T^a^ Max (°C)	R^a^ Max (%/°C)	T^b^ Onset (°C)	T^b^ Max (°C)	R^b^ Max (%/°C)	Residue (%)
Control sample	71.6	99.0	0.0161	307.7	343.2	2.0414	19.57
BDA-30	67.3	88.4	0.0550	267.4	285.8	2.3141	36.21
BDA-60	66.6	88.3	0.0516	269.1	285.3	2.7191	34.95
BDA-90	68.5	90.8	0.0556	270.6	287.4	2.4666	36.33
BDA-120	67.3	89.2	0.0516	268.5	285.1	2.6325	34.84
BDA	103.7	147.6	0.8335	252.6	260.0	1.9297	42.26

**Table 5 nanomaterials-12-04048-t005:** Relevant chemical bonding energies in BDA.

Chemical bond	P–C	C–N	P–N	C–O	C–C
Bond energy(kJ/mol)	264	293	300	326	335
Chemical bond	B–C	B–N	P–O	P=O	B–O
Bond energy(kJ/mol)	356	389	410	585	774

**Table 6 nanomaterials-12-04048-t006:** The tensile strength of control sample and BDA treated samples (3 curing times, 50 LCs).

Sample	Tensile Strengths (N)	The Loss Rate of Breaking Strength (%)
Control sample	530	-
BDA-30	472	10.94
BDA-60	455	14.15
BDA-90	409	22.83
BDA-120	393	25.85

**Table 7 nanomaterials-12-04048-t007:** The whiteness values treated samples with different laundering cycles (3 curing times).

LCs	Whiteness Value (Whiteness Ratio)			
	BDA-30	BDA-60	BDA-90	BDA-120
1	64.68 (85.4)	63.94 (84.4)	61.84 (81.7)	64.74 (85.5)
10	66.54 (87.9)	65.20 (86.1)	63.85 (84.3)	65.84 (86.9)
30	67.88 (89.6)	67.04 (88.5)	64.54 (85.2)	66.90 (88.3)
50	69.40 (91.6)	67.98 (89.8)	66.24 (87.5)	67.60 (89.3)

## Data Availability

The data presented in this study are available on request from the corresponding author.

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
