# Peer review of "Reactive Flame-Retardant Cotton Fabric Coating: Combustion Behavior, Durability, and Enhanced Retardant Mechanism with Ion Transfer"

_nanomaterials, 2022, doi:10.3390/nano12224048_

Round 1

Reviewer 1 Report

This paper is paper presented a lot of work; however, according to me, this paper is not suitable for this journal ' nanomaterials'. It could be suitable for some textile journal or MDPI Polymer journal in some special issues like ' Recent Advances in Textiles and Fibers'. 

Reviewer 2 Report

My comments are in attached file.

Reviewer 3 Report

The article about Reactive Flame-Retardant Corron Fabric coating is well written and can be of interest to the reader of nanomaterials but the authors need to address the following issues before it can be published. More in details:

Please add how many samples were analyzed for the FTIR analysis (and also an information about the reproducibility of the results obtained with the SEM images).

Line 145 check space in the equation

The language should be checked before the article will be published

Check page 1 line 42: “it still cannot compete” (retardants is plural)

Line 43 – 44  flame retardant efficiency of phosporus based flame retardants (repetition) etc

line 46  materials’ (?)

line 45-49 flame retardants is repeated 4 times in 4 lines

line 51-52 flame retardants is repeated other 4 times in 4 lines

etc

as a note (and also curiosity drive question) are 4 decimals place really necessary in the description of synthesis of BDA (see line 124 and boric acid (2.4732 g, 0.04 mol)) is such precision required? (also see After the addition of urea (12.012 g)) see also line 334  (Rbmax of 2.0414%)
